



# INFRASOUND MEASUREMENT SYSTEM FOR REAL-TIME IN-SITU TORNADO MEASUREMENTS

Brandon C. White, Brian R. Elbing, and Imraan A. Faruque

Oklahoma State University, Stillwater, Oklahoma, USA

**Correspondence:** Imraan A. Faruque (i.faruque@okstate.edu)

**Abstract.** Previous work suggests that acoustic waves at frequencies below human hearing (infrasound) are produced during tornadogenesis and continue through the life of a tornado, which have potential to locate and profile tornadic events and provide a range improvement relative to current radar capabilities, which are the current primary measurement tool. Confirming and identifying the fluid mechanism responsible for infrasonic production has been impeded by limited availability and quality

(propagation-related uncertainty) of tornadic infrasound data. This paper describes an effort to increase the number of measurements and reduce the uncertainty in subsequent analysis by equipping storm chasers and first responders in regular proximity to tornadoes with mobile infrasound measurement capabilities. The study focus is the design, calibration, deployment, and analysis of data collected by a Ground-based Local INfrasound Data Acquisition (GLINDA) system that collects and relays data from an infrasound microphone, GPS receiver, and an IMU. GLINDA has been deployed with storm chasers beginning in

May 2020 and has provided continuing real-time automated monitoring of spectrum and peak detection. In analysis of sampled severe weather phenomena, the signal measured from an EFU tornado (Lakin, KS) show an elevated broadband signal between 10 and 15 Hz. A significant hail event produced no significant increase infrasound signal despite rotation in the storm. The consistency of these observations with existing fixed array measurements and real-time tools to reduce measurement uncertainty demonstrates the value of acquiring tornado infrasound observations from mobile on-location systems and introduces a

capability for real-time processing and display of mobile infrasonic measurements.

## 1   Introduction

Tornadoes remain a significant hazard to life and property. In the United States, 800-1400 annually reported tornadoes claim an average of 55 lives (Ashley, 2007; Paul and Stimers, 2012) with 76 confirmed fatalities in the United States in 2020

(NOAA/SPC, 2021). Many of these fatalities occur in the southeast United States due, in part, to hilly terrain limiting line-of-sight measurements such as radar. This has motivated a search for alternative methods to complement radar measurements. Infrasound, sound below human hearing (< 20 Hz), is one possibility. Infrasound in the nominal range of 0.5 to 10 Hz has been observed coming from the same region as tornadoes that were verified with Doppler radar and/or visual observations (Bedard





et al., 2004b, a; Frazier et al., 2014; Goudeau et al., 2018; Elbing et al., 2019). These recent observations suggest that the infrasound signal may carry information specific to the tornado structure and dynamics. Since infrasound can be detected over long distances due to weak atmospheric absorption at these frequencies, it is a potential means of long-range, passive tornado monitoring.

The long propagation range of infrasound, coupled with the omnidirectional, continuous coverage provided by relatively inexpensive infrasound microphones could provide a significant improvement in our ability to detect, track, and ultimately predict and understand tornadic phenomena, as summarized in Table 1. However, the limited availability and quality of tornado-relevant infrasound measurement systems, coupled with the required development effort, has inhibited the integration of these measurements into a national framework. In this work, we introduce a portable infrasound measurement tool that may be carried by storm chasers and first responders, combined with a realtime interface portal. These are the first tools available to the public that are applicable to decentralized deployment that would provide widespread real-time infrasound coverage near tornado-bases without additional cost to the end user.

|  | Horizontal coverage | Temporal dynamics | Low altitude effect | Core physical effect |
|---|---|---|---|---|
| Tornado | localized | rapid | primary location | latent heat; wind |
| Radar | directional sweeps | periodic scan | ground clutter | moisture reflections |
| Infrasound | omnidirectional | continuous | reflections | pressure wave propagation |

**Table 1.** Infrasound has several potential advantages over existing weather radar measurements as a potential mechanism for tornado detection and tracking.

## 2 Previous Work and Background

The tornado-infrasound production hypothesis first appeared in 1960's conferences and only a minority of early work is available in archival journals (Georges, 1973). Contemporary tornado infrasound results continue to be reported primarily in conference papers (Noble and Tenney, 2003; Bedard et al., 2004b, a; Prassner and Noble, 2004), project reports (Rinehart, 2018), and oral presentations (Rinehart, 2012; Goudeau et al., 2018). Exceptions to this trend include four journal articles focused on infrasound observations from tornadoes (Bedard, 2005; Frazier et al., 2014; Dunn et al., 2016; Elbing et al., 2019). Bedard (2005) showed that infrasound emissions of ≈ 1 Hz followed the available radar observations associated with a tornado. Bedard (2005) also indicates that over 100 infrasound signals from the NOAA Infrasound Network (ISNET) were determined to be associated with tornado and tornado formation processes; the details of the association technique are not included in the article. Frazier et al. (2014) tracked tornadoes in Oklahoma using beamforming at infrasound frequencies. Dunn et al. (2016) detected a 0.94 Hz acoustic signature associated with an EF4 tornado in Arkansas using a ring laser interferometer.

Systematic infrasound observations from tornadoes and their formation processes remain a research challenge, which continues to contribute to large uncertainties associated with the observations and the underlying fluid mechanism responsible for its production (Petrin and Elbing, 2019). The need for detailed tornado related infrasound observations appropriate for archival





literature motivated an effort to increase both the number and quality of tornado infrasound observations, beginning with the 2016 installation of a fixed, 3-microphone infrasound array on the Oklahoma State University (OSU) campus. The OSU fixed array observations have provided some of the most recent and complete tornadic infrasound measurements, including signals associated with the 11 May 2017 EFU tornado (Elbing et al., 2019). The fixed nature of this array limited the proximity to tornadoes and radar sites, and though 2019 included a historically high number of tornadoes (NOAA, 2020c), no 2019 observa-

tions had both reliable infrasound and radar data. To address this gap, a mobile 4-microphone infrasound array was developed (Petrin et al., 2020), heliotrope solar hot air balloons (Bowman et al., 2020) were equipped with infrasound sensors (Vance et al., 2020), and a Ground-based Local INfrasound Data Acquisition (GLINDA) system was developed in 2019-20 to be carried by storm chasers and first responders. The result is a robust ecosystem of complementary measurement technologies that are capable of improving the number and quality of infrasound measurements near tornadoes.

This paper focuses on the design, deployment, and recent results from the mobile stormchasing unit (GLINDA). This unit provides a tool to mitigate uncertainties associated with atmospheric propagation by acquiring targeted close range measurements from verified tornadoes. The GLINDA unit was deployed with storm chasers beginning in the 2020 tornado season, and continues to provide valuable real-time measurements available through a web interface accessible to stormchasers and other meteorology partners.

This paper is organized as follows. Section 3 outlines the design goals, design process, and calibration procedure for the GLINDA unit. Section 4 covers the realtime processing needs and an automated analysis approach including variance reduction and peak-quantification. Section 5 describes two example storm measurements (a tornado and hail-producing storm) recorded by the unit during the 2020 storm season and used to verify its operation. Section 6 shows calibration, spectral, and peak identification results for the installed unit and the example storms.

## 3   GLINDA Design and Calibration

In this section, system design goals are identified, and hardware components, computational platforms, and data handling for collection and retention prior to analysis are discussed. Additionally, calibration procedures over the specific range of frequencies of interests are presented for the unit.

### 3.1   System Design

The primary measurement system goals were: (a) microphone signal resolution of 20mPa or better to provide comparable infrasound feature resolution to existing fixed arrays (Elbing et al., 2019), (b) positioning resolution under 10 m to provide comparable or better accuracy to current NOAA-reported tornado coordinates, and (c) "real-time" remote data relay, where real-time is defined as at or exceeding weather radar update frequencies (approximately once per 90 seconds).



### 3.1.1 Infrasound Microphone

GLINDA uses an off-the-shelf infrasound microphone (Model 24, Chaparral Physics), the same model used for the fixed 3-microphone array near Oklahoma State University (Elbing et al., 2019). The microphone has a sensitivity of 401 mV/Pa at 1 Hz and a flat response from 0.1-200 Hz to within -3 dB (flat to within -0.5 dB for 0.3 to 50 Hz). The typical frequency response for this model of microphone is provided in Fig. 1. The noise for this microphone at 1 Hz was -81.6 dB relative to 1 Pa$^2$/Hz. The microphone was mounted on the floorboard of the storm chasing truck as shown in Fig. 2. Concerns about the

ability to suppress wind noise motivated the mounting inside of the truck cab and, consequently, no additional windscreen was implemented. The maximum signal output for the microphone was 36 V peak-to-peak, and the analog output signal was sampled at 2050 Hz. By sampling at greater than 2000 Hz, GLINDA avoids aliasing in the frequency band of interest.

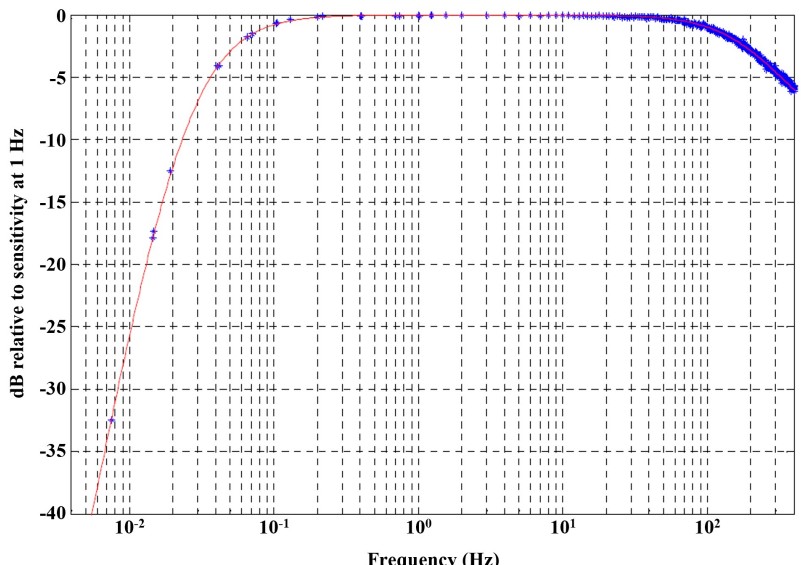

**Figure 1.** Typical frequency response curve for a Model 24 (Chaparral Physics) microphone, which was used for this deployment of GLINDA. Credit: Chapparal Physics.

### 3.1.2 Sensor Components

In addition to the infrasound microphone, GLINDA monitors and records data from an inertial measurement unit (IMU), a
Global Positioning System (GPS) receiver, and the analog-to-digital converter (ADC) that was connected to the analog output from the infrasound microphone. An illustration of data flow for the upload process from each of the sensors is provided in Fig. 3. The IMU, GPS, and ADC were sampled at 100 Hz, 1 Hz, and 2050 Hz, respectively. The microphone ADC (ADS1115, Adafruit Industries) has a 16-bit resolution and signal voltage tolerance of $\pm 5$V, giving a quantization increment of $76\mu$V. Since the microphone outputs a differential signal reading of 0-36 V, a voltage divider circuit was implemented with gain ratio





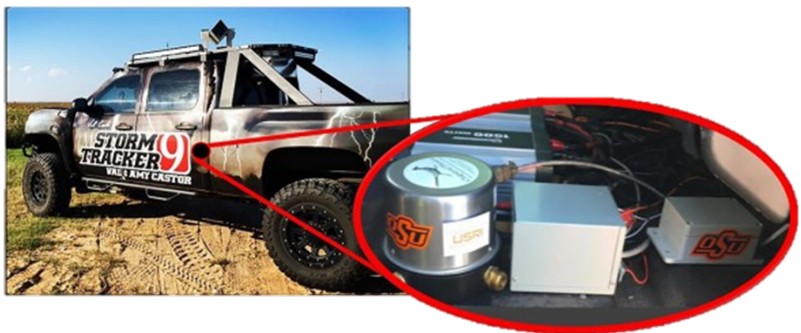

**Figure 2.** Image of GLINDA-housing storm chasing vehicle showing approximate location and configuration of install.

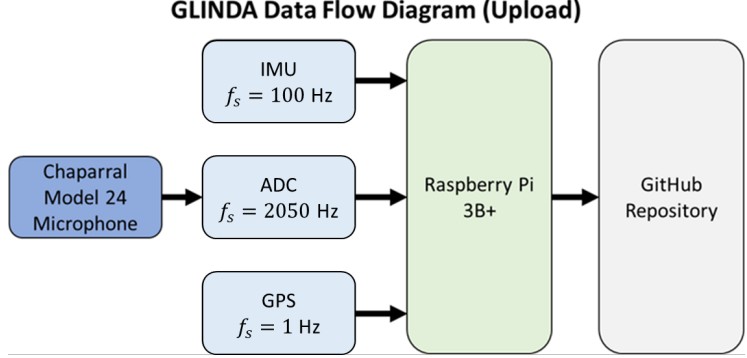

**Figure 3.** Flowchart for acquiring and uploading of data from GLINDA.

of 13.6% to ensure the measured signals fall within the ADC tolerance. The nominal microphone sensitivity of 400 mV/Pa provides a pressure resolution of 1.4 mPa, keeping GLINDA's quantization error below the existing fixed array's typical noise floor of 20 mPa observed during severe storms with high winds (Elbing et al., 2019). The GPS unit (746 Ultimate GPS Breakout Ver 3, Adafruit Industries) provides accuracy of 1.8 m and 0.1 m/s. 1.8 m radially is accurate to approximately 0.00001° for data collection latitudes. For comparison, current NOAA tornado reports typically provide GPS coordinates with three to four

decimals, and the system positioning accuracy exceeds these estimates. Each measurement is time-stamped and stored locally until export processes are initiated.

### 3.1.3    Computing Platform

The mobile GLINDA hardware is built on a Raspberry Pi 3B+ platform with a Raspbian distribution. The scripts were written in Python 3.7 and implement Adafruit libraries for reading the attached sensor packages over $I^2C$ and serial/UART connections.

Upon startup, the system initializes all sensors for data collection and continues recording until loss of power. Data is saved every 10 s to ensure minimal loss in the event of unexpected shutdown. Each data file is timestamped by the local system time





coordinated via Network Time Protocol thorough the Pi board. A system service initializes and maintains data uploads to a GitHub repository.

### 3.1.4   Installation and Deployment

For the 2020 tornado season, GLINDA measured from within the cab of a storm chasing truck operated by Val and Amy Castor (see Fig. 2), which provided live coverage of severe storms for a local news station in Oklahoma City, OK (News 9). GLINDA was installed during the first week of May 2020. It was powered by 1500 W inverters (Strongway), which also power the vehicle's other weather observation systems. The 60 Hz signal produced by the inverters is outside of the frequency band of interest. A small port in the roof of the vehicle allowed the GPS antenna to be routed outside of the cab for improved
connectivity. Data size is limited by a power switch integrated into the vehicle dash.

### 3.1.5   Data Recovery

An illustration of the data flow from download through processing is provided in Fig. 4. GLINDA maintains an internet connection via a router in the storm chasing vehicle that is primarily used to provide live video and audio to the local news outlet during storm chases. While connected to the internet, GLINDA scans the data storage directory for new or modified
files resulting from sampling of the sensor packages. An upload commit is generated to push new or updated files to the online repository (GitHub). File uploads for the system are conducted at regular intervals, typically at $\Delta t < 60$ s. Further discussion on selection for timing considerations is provided in section 3.1.

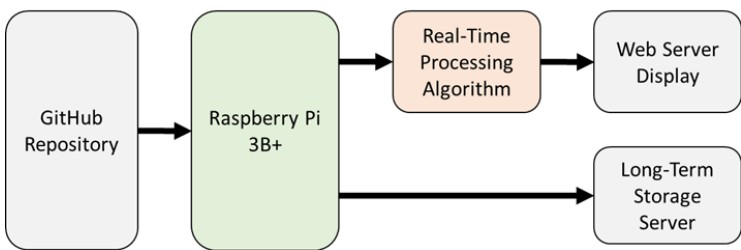

**Figure 4.** Data flow diagram for processing, display, and storage.

### 3.2   Calibration

To create a broadband acoustic and infrasound signal with sufficient excitation energy over the frequency bandwidth of in-
terest, the GLINDA system was calibrated utilizing a series of impulsive acoustic signals generated from a 20-gauge shotgun discharged at $< 2$ ft range and oriented approximately 120 degrees from the unit installed in the vehicle model cabin (2019 Ford F-150). A gunshot was used to generate the impulsive signals of the test to generate significant acoustic and infrasonic spectra with similar magnitude. Two series of three gunshots were completed in cabin configurations of windows up and

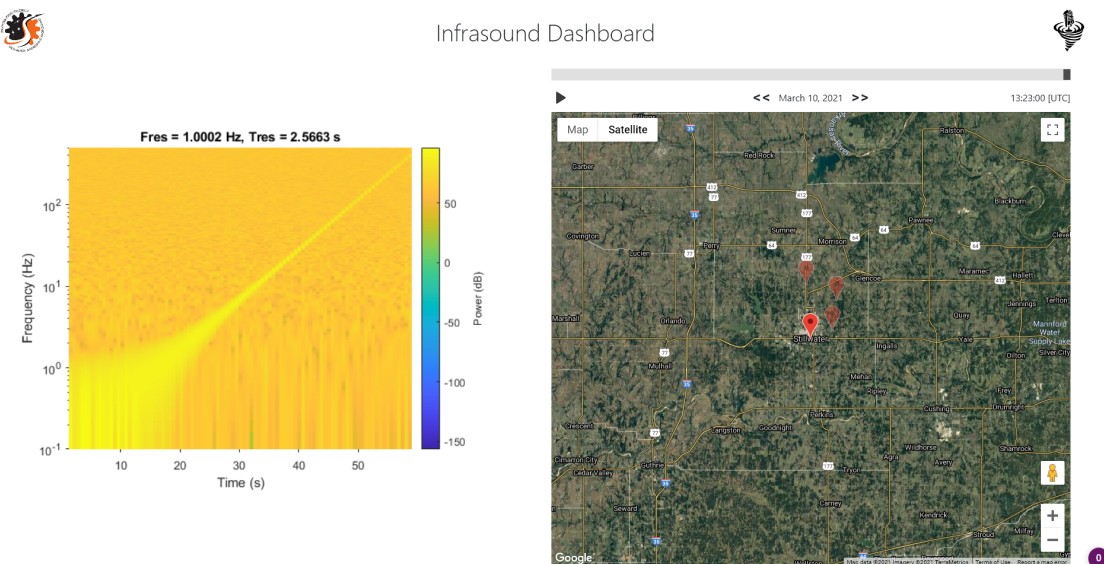

**Figure 5.** Example visualization of GLINDA-monitoring web interface (see map graphic for copyright).

down. The gunshots from each configuration were then compiled into a single data series for frequency domain analysis us-
ing approximately 1 s sections of recording containing each shot (approximately 0.1 s ringing) was utilized. By comparing
the signal measured outside of the vehicle to the interior microphone signal, an experimental transfer function was identified
for the acoustic impact of the vehicle over the frequency band of interest. A frequency domain model fit to the magnitude
measurements was used as the transfer functions for the window up and down cases.

### 3.3   Real-time display interface

To provide continuous monitoring and rapidly identify temporal trends in acoustic measurements allowing storm chasers to in-
clude infrasound signals in tactical decision-making, a web interface was developed that displays the GLINDA measurements
in near real time. The primary visualizations of the data are a spectrogram displaying near real time frequency decompositions
and a maps API displaying the location of the storm chasing unit via GPS. Figure 5 shows a demonstration of these visual-
ization capabilities with simulated inputs. In the spectrogram, $F_{res}$ and $T_{res}$ represent the size of the frequency and time bins
respectively. In addition to visualizing current measurements, a slider for time and date allows browsing historical data.

## 4   Analysis approach

In this section, a real-time analysis approach is developed that provides high-resolution spectral measurements over the region
of interest, and provides robust peak-detection and finding routines.





| Radar Measurement Method | Update Time (s) |
|---|---|
| Full Volume Scan - WSR-88D | 270 |
| $0.5^o$ Scan, SAILS(x1) - WSR-88D | 147 |
| $0.5^o$ Scan, SAILS(x2) - WSR-88D | 108 |
| $0.5^o$ Scan, SAILS(x3) - WSR-88D | 89 |

**Table 2.** Average minimum wait time between low-inclination update scans for varieties of WSR-88D scanning methods.

### 4.1 Real-Time Processing Qualifications

Traditional weather monitoring of severe weather, such as tornadoes in the United States, is completed via radar analysis. The United States National Weather Service (NWS) utilizes a wide-spread network of weather radars (most notably Weather Surveillance Radar, 1988 Doppler or WSR-88D) to provide the most accurate and frequent images available (US-NWS). Development of scanning methods that decrease time intervals between low-angle atmospheric sweeps for the WSR-88D have been the subject of repeated study and implementation (Daniel et al., 2014). The current methodology for minimizing the interval

between scans is the Multiple Elevation Scan Option Supplemental Adaptive Intra-Volume Low-Level Scan or MESO-SAILS. The average time between MESO-SAILS scans at the 0.5 degree radar inclination is provided in Table 2 and implies that GLINDA should target measurement intervals of $\Delta T_m \leq 90$ s to be comparable with current radar technology.

The $> 90$sec update rate, data generation rate, and desired bandwidth usage is used to identify the maximum length of an

individual recorded segment that keeps pace with radar as

$$89 \geq \Delta T_t = \Delta T_m + \Delta T_u = \frac{R_G \Delta T_m}{0.1 R_u} + \Delta T_m, \tag{1}$$

where $R_G$ is the rate of sensor data production, $R_u$ is the minumum connection speed expected, total time between measurements ($\Delta T_t$) is the sum of the time interval of data collection and the time required to upload the data ($\Delta T_u$), and GLINDA data usage is limited to less than 10% of the available bandwidth. For the GLINDA unit, $R_G = 70$ Kb/s and the upload constraint

is $1.35 \Delta T_m \leq 89$ seconds.

### 4.2 Spectral transformation

In 2020, GLINDA recorded several chases during severe weather events, including a dust storm, gustnado, and significant hail events. This paper analyzes the data acquired during a tornado-producing supercell including tornadogensis and a severe hail event without tornadic activity. Traditionally the frequency decomposition of a time-domain signal is performed using a Fast

Fourier transform (FFT) which returns a frequency domain representation of the data with linearly-spaced frequency points over the frequency band $f_{FFT} \in [0, f_s/2]$, where $f_s/2$ is commonly known as the Nyquist frequency. For the oversampled case, a large number of these points would be outside the frequency band of interest. In this study, we used the oversampling to reduce the frequency domain error by implementing a Chirp-Z transform (CZT) to allow the frequency domain resolution to be





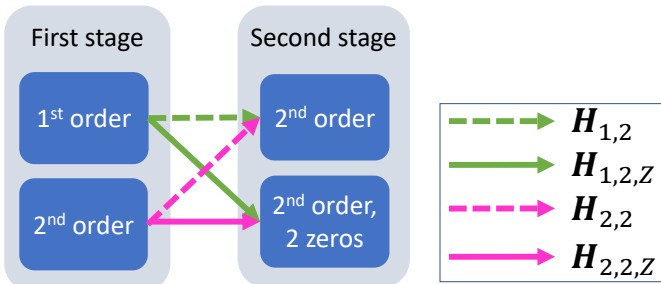

**Figure 6.** Frequency domain model structures considered and two-stage model fitting process.

directed only across a desired frequency band (Rabiner et al., June 1969), which is advantageous because the frequency band
of interest for the current work is a relatively small fraction of the band returned by the FFT ($\leq 10\%$). Thus the CZT produces
higher resolution over the desired range relative to an FFT. The CZT also has the advantage of reducing the required processing
time given the narrowed band, which is critical for enabling real-time analysis of infrasound measurements. However, due to
the data sizes presented in the current study the CZT and FFT had negligible runtime differences. The CZT, defined in Eq. (2),
takes a time domain series of $N$ points, $x(n)$, and transforms it into the complex Z-domain at a finite number of points along
a defined spiral contour $z(k)$ returning frequency domain signal, $X(k)$. Here $z(k)$ is a function of a complex starting point $A$,
the complex ratio between points $W$, and the number of spiral contour points $M$. For storm analyses over expected frequency
bands previously associated with tornadic acoustics, a complex spiral was defined as given in Eq. (3), which corresponds to a
band of 1-250 Hz with a frequency resolution of $\Delta f = 0.125$ Hz. A ten minute selection of microphone data was selected from
one hour before the event, spanning the event, and one hour after the event for each case presented. The ten minute intervals
were windowed using Hanning windows with 60% overlap and segmented into 15 s lengths.

$$X(k) = \sum_{n=0}^{N-1} x(n)z(k)^{-n}, \, z(k) = AW^{-k}, k = 0, 1, \ldots, M-1 \tag{2}$$

$$A = e^{2\pi i/1000}, \, W = e^{(2\pi/1000)(249/2000)i}, \, M = 2000 \tag{3}$$

### 4.3   Peak identification

To robustly identify the frequency domain peak in realtime, a model-based approach was taken, using a two-stage process to
compare fits to the four frequency domain model structures shown in Fig. 6. Each fit was derived by minimizing the mean-



squared error (MSE) between the measured SPL ($\hat{Y}$) and frequency response magnitude of the transfer function ($|H|$) as Eq 4.

$$MSE = \sum_{f=0.1\,\text{Hz}}^{100\,\text{Hz}} (\hat{Y}(f) - |H|(f))^2 \tag{4}$$

## 5  Observations

In this section, two significant weather events measured by GLINDA during the Spring 2020 tornado season and selected for detailed analysis in this study are described.

### 5.1  Tornado Event (22 May 2020)

A cold front that pushed in from the northwest of Kansas late on 21 May 2020 and into the early hours of 22 May 2020 produced several severe storms. One storm, pictured in Fig. 7, produced a tornado near Lakin, Kansas. The tornado touchdown

at 0011 UTC at coordinates (37.802, -101.468) and ended at 0024 UTC at coordinates (37.7982, -101.4387). It was 2.83 km (1.76 miles) in length, had a maximum damage path width of 137 m (150 yards), and was classified as an EFU by the NWS because it tracked over an open field that produced insufficient recorded damage for reliable categorization (NOAA, 2020b). The storm chasers, equipped with GLINDA, arrived to the intercepting location for the tornadic storm system approximately 2-5 minutes prior to tornadogenesis. The intercepting storm chasers were located approximately 4 km SSE of the tornado

during tornadogenesis. Local radar and radial velocity data for this storm are provided in Appendix B.

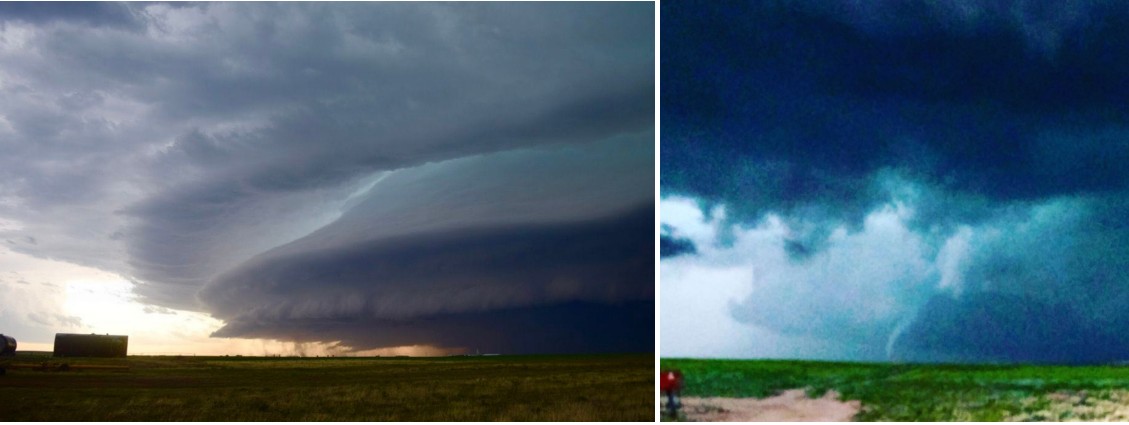

**Figure 7.** (Left) Picture of the storm system that produce the tornado near Lakin, KS on 22 May 2020. (Right) Picture of the Lakin, KS tornado. Photo credit: Val and Amy Castor.





## 5.2  Large Hail (22 May 2020)

The day after the Lakin EFU tornado, a stalled outflow boundary intersected a dryline resulting in the firing of numerous severe thunderstorms. One of these large supercell storms produced baseball-sized hail as it moved over southern Oklahoma. The GLINDA equipped storm chasers intercepted the storm as it moved through Comanche County and produced 25.4 mm (1 inch) hail at 2315 UTC on 22 May 2020 near the coordinates (34.62, -98.75) (NOAA, 2020a). Of note, this storm did exhibit weak rotation at times, but a tornado was never produced.

## 6  Results & Discussion

In this section, the calibration results are presented, the frequency domain analysis of the two storms are compared, and the peak finding routine is applied to signals with an infrasound rise. The spectral comparison shows a rise in infrasonic signals in the presence of a tornado-producing supercell near tornadogenesis. The rise is statistically significant with respect to the signal noise and standard deviation, and does not appear after the tornado or in a non-tornadic hail storm which contained rotation.

### 6.1  Calibration

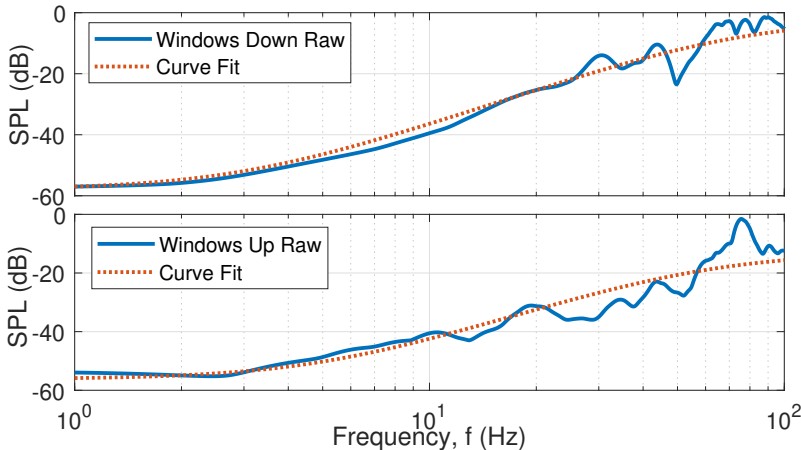

**Figure 8.** Experimental transfer function data and fitted models for windows up and windows down configurations.

Figure 9 provides the measured frequency response as obtained through calibration described in section 3.2 for a windows up and down configuration. A frequency-domain model fit to the magnitude measurements then gives transfer functions $H_u(s)$ and $H_d(s)$ for the window up and down cases (Eq. 5 and 6), respectively. These transfer functions are compared in Fig. 9.

$$H_u = \frac{0.22(s + 31.5)^2}{(s + 377)^2} \tag{5}$$





$$H_d = \frac{0.79(s+18.8)^2}{(s+471)^2} \qquad (6)$$

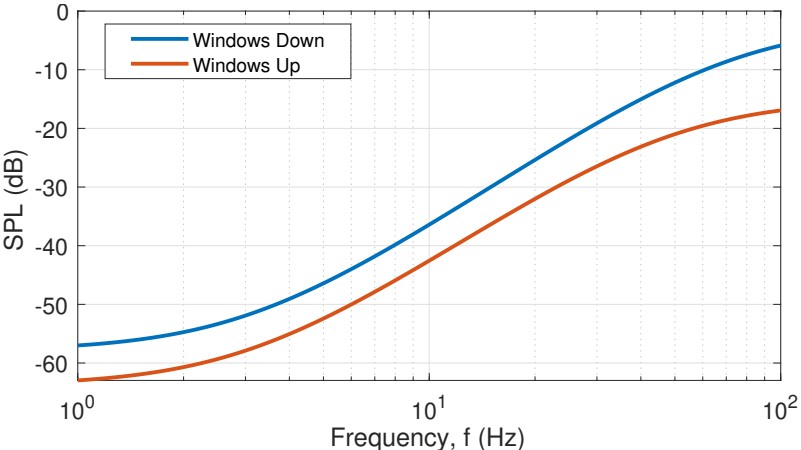

**Figure 9.** Comparison of models for windows up and windows down configurations

The calibration was conducted in a vehicle example for both windows up and down. The window position may have varied during storm chasing events, and the spectral analysis and peak finding algorithm in sections 6.2 and 6.3 are discussed relative to the recorded signal (as implemented in the real-time workflow) and only a single unit is analyzed. For analysis of the underlying physical mechanisms responsible for tornado infrasound or for comparison between multiple units, the truck's acoustic response may be removed by applying the inverse of the calibration transfer function.

## 6.2   Spectral results

The spectral content recorded on GLINDA before, during, and after the Lakin tornado are compared in Fig. 12. To reduce the effect of the significant spectral slope, a linear regression was computed as SPL change (dB) per frequency decade for pre-, post-, and during event conditions as defined in (7) and over the frequency range of [0.1 250] Hz. The trendlines have slopes within 0.5 dB/decade of each other as shown in Fig. 11, while the varied 1 Hz intercepts also suggest an overall rise in spectral energy content across the band of interest during tornado interception. Acoustic work often associates broadband SPL rise below 500Hz with wind (Nelke et al., 2016; Lin et al., 2014; Nelke et al., 2014), which does not necessarily extrapolate to the sub-acoustic region under consideration here.

$$\hat{X}_{SPL}(f) = a_0 \log_{10}(f) + a_1 \qquad (7)$$

The variance about the best fit curve for each case illustrates the noise reduction in frequency domain due to windowed averaging. The raw and windowed variances for the pre-, post-, and during tornado intervals are provided in Fig. 10. This





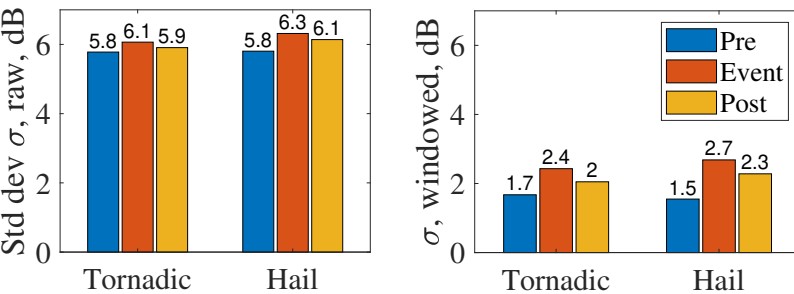

**Figure 10.** Standard deviation of spectra.

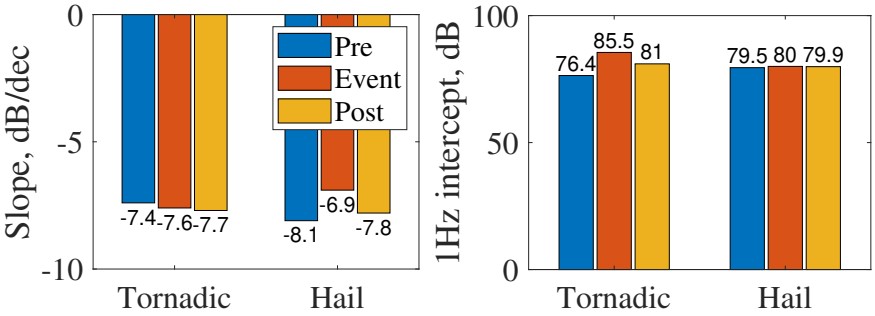

**Figure 11.** Linear fit parameters.

comparison shows that the frequency-domain averaging provided by windowing reduced the numerical variance from 33-35

235   dB in the raw transform to 2.9-5.9 dB variance in the windowed transform. This noise reduction significantly improves the

ability to resolve features with changes in the 3-6 dB magnitude range. From this variance reduction, the elevated signal in the

10 to 15 Hz frequency band during the tornado is made more apparent. The center of this frequency band (10-15 Hz) shows a

9 dB rise above the linear fit, and this rise over this frequency band is not present one-hour prior to or one-hour following the

tornadic activity.

The 10-15 Hz elevated frequency band during the tornado accounts for 3.3 dB (and therefore the majority) of the variance

during the tornadic event and the peak falling within this range is consistent with this being a relatively small tornado. Tornado

infrasound is consistently reported with a fundamental frequency in the 0.5 to 10 Hz range (Bedard, 2005) with the smaller

the tornado the higher the frequency. Elbing et al. (2019) observed a similar small EFU tornado and its fundamental frequency

was estimated to be 8.3 Hz. These observations are nominally consistent with the analysis of (Abdullah, 1966) that predicts

$f_n = (4n + 5)c/4d$, where $f_n$ is the frequency of mode $n$, $c$ is the speed of sound, and $d$ is the diameter of the vortex core.

There are fundamental issues with this analysis, but all of the results published in archival journals (Bedard, 2005; Frazier

et al., 2014; Dunn et al., 2016; Elbing et al., 2019) nominally follow this trend (though generally aligning better with the first

overtone, $n = 1$). Thus it is appropriate to use this relationship as a nominal empirical relationship. This analysis predicts that

a tornado with a fundamental frequency of 12.5 Hz would have a vortex core diameter of 34 m. Past observations indicate that





this estimate is likely to be low and the actual tornado core that would produce a 12.5 Hz fundamental frequency would fall in the range of 35-90 m. This is still below the reported maximum damage path width (137 m). However, having a similar magnitude is likely all that can be expected given the uncertainty in the analysis, damage assessment (both estimated values as well as only reporting the maximum), and relationship between vortex core and damage width. Note that the largest tornadoes have core diameters well in excess of 1000 m.

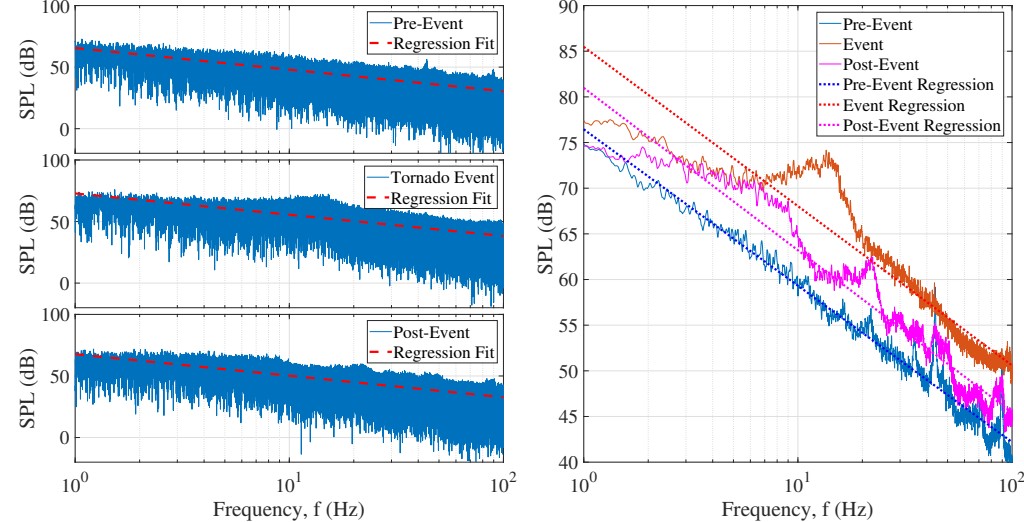

**Figure 12.** Frequency domain GLINDA data during the Lakin, KS tornadic event that occurred from 0011 UTC to 0024UTC on 22 May 2020.

The spectral content associated with the Commanche County hail event is shown in Fig. 13. As with the Lakin tornado analysis, calculation of best fit lines (defined in (7) with fitting parameters listed in Fig. 11) show over 30 dB reduction for windowed spectra as listed in Fig. 10. The linear best fit lines for the hail event do not show significant SPL differentials between the three intervals (pre-, post-, and during the hail) at the lower end of the measurement range $< 10$ Hz. The slope of the fit during the hail event is 3 dB/decade higher than the pre- or post-events which creates a SPL difference of up to 5 dB

over the band of interest. Unlike the Lakin EFU tornado, there was no apparent swell in SPL in the 10 to 15 Hz band. However, smaller rises near 50 and 80 Hz for the event spectra are present with a peak of 6 dB relative to the linear best fit. It is unclear what is responsible for these peaks, but these features are at frequencies above what is typically associated with severe weather (though some overtones have produced signals in the audible range). In the frequency range of interest (nominally 1-10 Hz), there is no apparent signal that was produced. This is consistent with past observations that have noted that hail producing

storms without tight rotation typically do not produce an infrasound signal (Petrin and Elbing, 2019).





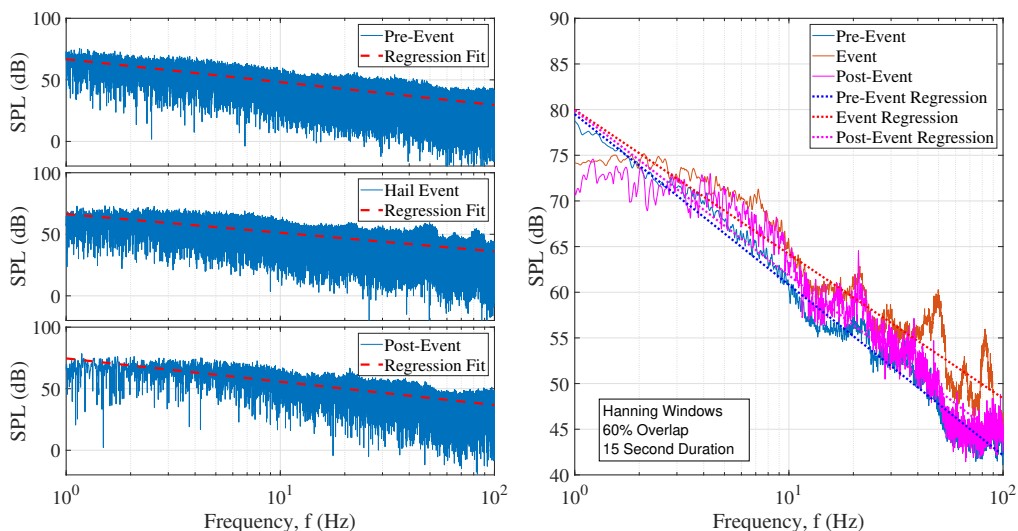

**Figure 13.** Frequency domain infrasound measured with GLINDA during the Comanche County hail event at 2315 UTC on 22 May 2020.

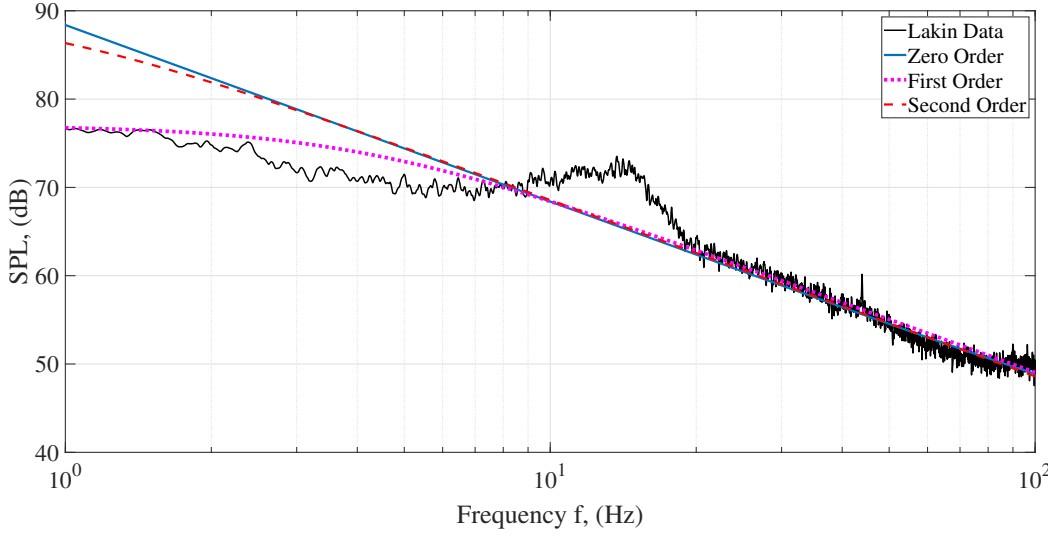

**Figure 14.** First stage model fit to Lakin, KS tornado infrasound/low-sonic measurements.

## 6.3 Peak identification results

For the tornadic measurements, an infrasound peak was present and the results of the two-stage model fitting process in Fig. 6 are presented in Tables 3 and 5.

From the first stage model fits, a first order fit has the lowest mean-squared error (MSE) between the tested transfer function models. The first order model and second order model were further investigated through a second stage transfer function model






| TF Form | MSE (dB$^2$) |
|---|---|
| $k_0 + \dfrac{a_1}{s}$ | 6.98 |
| $\dfrac{a_0}{s + a_1}$ | 3.28 |
| $\dfrac{a_0}{s^2 + a_1 s + a_2}$ | 5.31 |
| $\dfrac{a_0}{s^3 + a_1 s^2 + a_2 s + a_3}$ | (Did not converge) |

**Table 3.** Mean-Squared Error for various first stage curve fits to raw data.

fitting. To develop the second grouping of models, the difference between the raw data and the first stage models was compared to an more select group of transfer functions based on observable characteristics in the resulting difference signal - namely the rise over the 10-15 Hz band and the flat region in the audible range. Results of the overall model fits comprised by adding the fitted models from the secondary analysis to the first stage model fit are presented in Table 4 where values of the percent errors

for detected frequency for the peak (Eq 8) and detected peak magnitude (Eq 9) were calculated relative to manually gathered values. The MSE for the combined fits is additionally presented.

$$E_{f_p} = |\frac{f_p - \hat{f}_p}{\hat{f}_p}| \tag{8}$$

$$E_{|H|_p} = |\frac{|H|_p - \hat{Y}_p}{\hat{Y}_p}| \tag{9}$$

Although $H_{2,2,Z}$ has the lowest overall MSE and percent error for peak magnitude, the coefficients returned by the optimization process are of large order and have significantly larger adjacency value bounds as compared to first stage, first order fits (Table 5), creating over-fitting concerns. $H_{1,2,z}$ by contrast does not produce similarly large coefficients and low adjacent value bounds. This is achieved while only producing a 3 percentage point increase in the peak percent error and 0.8 dB$^2$ mean square error over the frequency band suggesting it may be a more viable model candidate. An additional round of fits was completed

for the $H_{1,2,Z}$ model structure implementing a log frequency weighting to the mean-squared error formulation in 4 as a decadal Gaussian distribution centered on $f = 10$ Hz. The weighting scheme decreased $E_{|H|_p}$ by 12% in exchange for $f_p$ and $MSE$ increasing 3%. Because the focus of this modelling approach is largely concerned with peak frequency identification, the usage of this weighting method was not further implemented for usage.

The model structure fitting process is a nonlinear minimization that is prone to local minima or poorly fit terms. To examine

the robustness of the model fits to differing initial guesses, a Monte-Carlo approach to fitting model uncertainty was implemented by taking one thousand initial parameter guesses uniformly distributed over one order of magnitude surrounding what is an expected range of values.





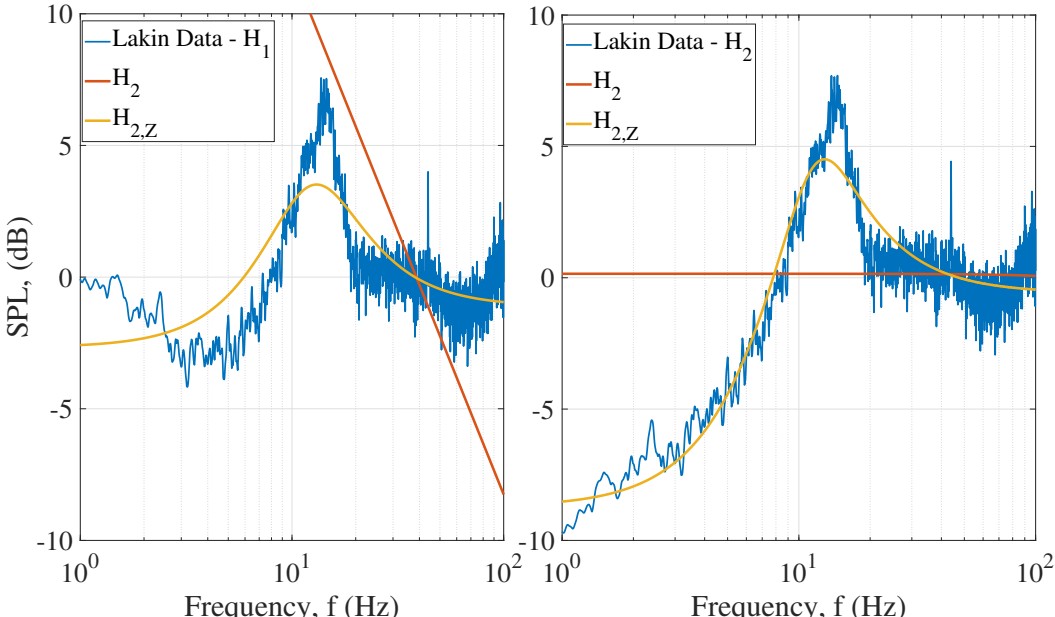

**Figure 15.** Secondary curve fits to Lakin, KS tornado measurements with select first stage fits subtracted.

| Fit Name | Stage 1 | Stage 2 | $\mathbf{E_{f_{peak}}}$ (%) | $\mathbf{E_{|H|_{peak}}}$ (%) | MSE (dB$^2$) |
|---|---|---|---|---|---|
| $\mathbf{H_{1,2}}$ | $\dfrac{a_0}{s+a_1}$ | $\dfrac{b_0}{s^2+b_1 s+b_2}$ | 99.3 | 111.4 | 64.9 |
| $\mathbf{H_{1,2,Z}}$ | $\dfrac{a_0}{s+a_1}$ | $b_0\dfrac{(s+b_1)^2}{s^2+b_2 s+b_3}$ | 6.6 | 8 | 1.82 |
| $\mathbf{H_{2,2}}$ | $\dfrac{a_0}{s^2+a_1 s+a_2}$ | $\dfrac{b_0}{s^2+b_1 s+b_2}$ | 99.3 | 45.9 | 5.32 |
| $\mathbf{H_{2,2,Z}}$ | $\dfrac{a_0}{s^2+a_1 s+a_2}$ | $b_0\dfrac{(s+b_1)^2}{s^2+b_2 s+b_3}$ | 8.2 | 5.3 | 1.08 |

**Table 4.** Fit parameters for the four model structures.

| | | $a_0$ | $a_1$ | $a_2$ | $b_0$ | $b_1$ | $b_2$ | $b_3$ |
|---|---|---|---|---|---|---|---|---|
| $H_{1,2}$ | Median | $1.78 \times 10^5$ | 25.15 | – | $1.43 \times 10^5$ | 549 | $2.5 \times 10^4$ | – |
| | Adjacent Values | $\pm 10^{-4}\%$ | $\pm 4 \times 10^{-5}\%$ | – | $\pm 10^{-4}\%$ | $\pm 10^{-4}\%$ | $\pm 4 \times 10^{-4}\%$ | – |
| $H_{1,2,Z}$ | Median | $1.78 \times 10^5$ | 25.15 | – | 0.879 | 72.411 | 85.408 | 6266 |
| | Adjacent Values | $\pm 3 \times 10^{-6}\%$ | $\pm 3 \times 10^{-5}\%$ | – | $\pm 3 \times 10^{-6}\%$ | $\pm 3 \times 10^{-5}\%$ | $\pm 4 \times 10^{-5}\%$ | $\pm 10^{-4}\%$ |
| $H_{2,2}$ | Median | $8.53 \times 10^{12}$ | $5.4 \times 10^7$ | $4.1 \times 10^5$ | $1.3 \times 10^5$ | 464 | $2.5 \times 10^4$ | – |
| | Adjacent Values | $\pm 450\%$ | $\pm 415\%$ | $\pm 5.5 \times 10^3\%$ | $\pm 39.9\%$ | $\pm 58.4\%$ | $\pm 4 \times 10^{-4}\%$ | – |
| $H_{2,2,Z}$ | Median | $9.2 \times 10^{12}$ | $4.7 \times 10^7$ | $2.5 \times 10^4$ | 0.934 | 45 | 56.2 | 5529 |
| | Adjacent Values | $\pm 386\%$ | $\pm 466\%$ | $\pm 7.9 \times 10^5\%$ | $\pm 9.1\%$ | $\pm 113\%$ | $\pm 95\%$ | $4.6\%$ |

**Table 5.** Median fit parameters uncertainty distribution as quantified by Monte Carlo convergence over one thousand initial guesses.





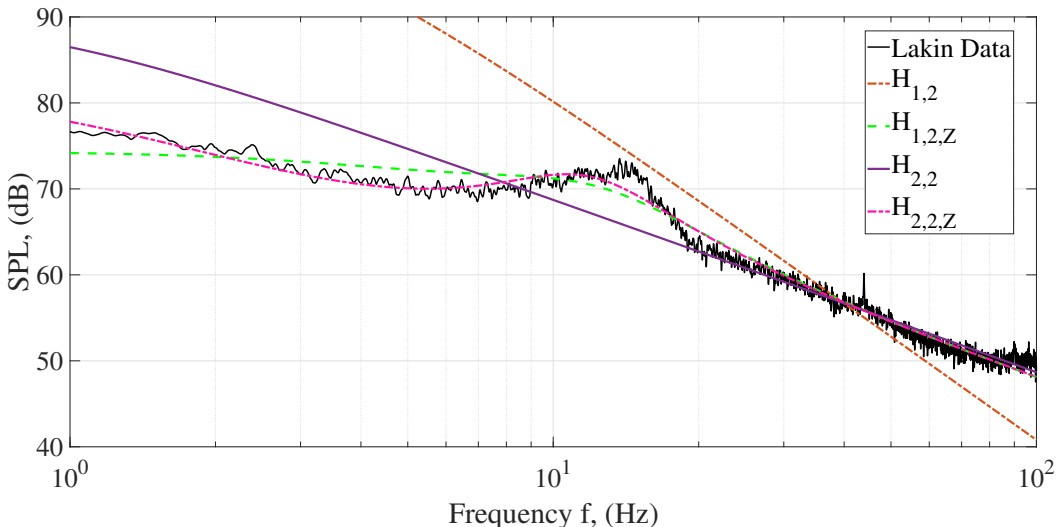

**Figure 16.** Comparison of frequency-domain model fits to Lakin, KS tornado measurements.

## 7   Conclusions

Sparse tornadic infrasound measurements have limited their application in predictive work. To improve the availability and quality of observations of infrasound from tornadoes, the Ground-based Local INfrasound Data Acquisition (GLINDA) system that includes an infrasound microphone, an IMU, and a GPS receiver has been designed and deployed. The infrasound microphone has sensitivity of 0.401 V/Pa with a flat (to -3 dB) response from 0.1 to 200 Hz and the 36 V peak-to-peak analog output is sampled at 2050 Hz via ADC to mitigate aliasing of signals with significant magnitude. This microphone, voltage divider circuit, and ADC measurement path maintains resolution of 1.4 mPa which sets the quantization error below existing noise floor estimates from existing array measurements at 20 mPa. The GLINDA system operates on a Raspberry Pi 3B+ platform with the data exported to an online repository and processed for realtime display of spectra and model fitting in an online display portal with update rates similar to radar. The processing tools incorporate the Chirp Z transform and windowing to reduce the uncertainty of the signal, as verified by a 50% reduction in standard deviation.

GLINDA was installed in a storm chasing vehicle and has been acquiring data from May 2020 to present. It has recorded several severe weather events, including a dust storm, gustnado, fires, and significant hail. This paper provides design details and analysis from two events - a tornado and a significant hail storm. The tornado occurred in the early hours of 22 May 2020 near Lakin, KS. The tornado lasted  13 minutes, had a length of 2.8 km, and a maximum damage path width of 137 m, and unknown strength. The storm chasers measured this system from 4 km SSE of the tornado. The spectral content shows an elevated signal during the tornado spanning 10 to 15 Hz, consistent with past observations of small tornadoes as described in section 2. The hail event occurred the following evening at 2315 UTC on 22 May 2020 with 25.4 mm hail while the storm





chasers were located in Oklahoma's Comanche County. Similar spectral analysis was performed on this event but no significant infrasound production was identified relative to periods before and after the hail.

These results indicate consistency of the mobile observations with fixed measurements and support this modality as a means of increasing the availability and signal to noise ratio of tornado infrasound observations, and the analysis shows an improvement in precision enabled my real-time model-fitting based processing tools to resolve and quantify spectral deviations associated with tornado activity.

*Data availability.* Realtime data portal access will be available to weather partners by request at www.autophysics.net.



## Appendix A: NWS WSR-88D Scanning frequency

### Table 1: SAILS Test VCP Definitions

| Elevation Angles (VCP 12) | VCP 12 Elevation Duration | SAILS | SAILSx2 | SAILSx3 |
|---|---|---|---|---|
| 0.5° | 31 Sec | 31 Sec | 31 Sec | 31 Sec |
| 0.9° | 31 Sec | 31 Sec | 31 Sec | 31 Sec |
| 1.3° | 31 Sec | 31 Sec | 31 Sec | 31 Sec |
| 0.5° | | | | 31 Sec |
| 1.8° | 15 Sec | 15 Sec | 15 Sec | 15 Sec |
| 0.5° | | | 31 Sec | |
| 2.4° | 14 Sec | 14 Sec | 14 Sec | 14 Sec |
| 3.1° | 14 Sec | 14 Sec | 14 Sec | 14 Sec |
| 0.5° | | 31 Sec | | |
| 4.0° | 14 Sec | 14 Sec | 14 Sec | 14 Sec |
| 0.5° | | | | 31 Sec |
| 5.1° | 14 Sec | 14 Sec | 14 Sec | 14 Sec |
| 6.4° | 14 Sec | 14 Sec | 14 Sec | 14 Sec |
| 0.5° | | | 31 Sec | |
| 8.0° | 13 Sec | 13 Sec | 13 Sec | 13 Sec |
| 0.5° | | | | 31 Sec |
| 10.0° | 13 Sec | 13 Sec | 13 Sec | 13 Sec |
| 12.5° | 13 Sec | 13 Sec | 13 Sec | 13 Sec |
| 15.6° | 13 Sec | 13 Sec | 13 Sec | 13 Sec |
| 19.5° | 13 Sec | 13 Sec | 13 Sec | 13 Sec |
| Duration | 243 Sec | 274 Sec | 305 Sec | 336 Sec |
| 0.5 Elevation Update Times | 253 Sec* | 136 Sec, and 148 Sec* | 108 Sec, 101 Sec and 106 Sec* | 93 Sec, 88 Sec, 72 Sec and 93 Sec* |
| | | Avg 147** Sec | Avg 108** Sec | Avg 89** Sec |

* 10 seconds were added to account for Retrace Time   ** The Avg estimate includes 20 seconds to account for Retrace and Elevation Transition times

**Figure A1.** WSR-88D MESO-SAILS scanning regiment with average times for measurement sweeps. From NOAA (2014)





**Appendix B: Tornado Event Auxiliary Data**

**B1 Reflectivity Measurements**

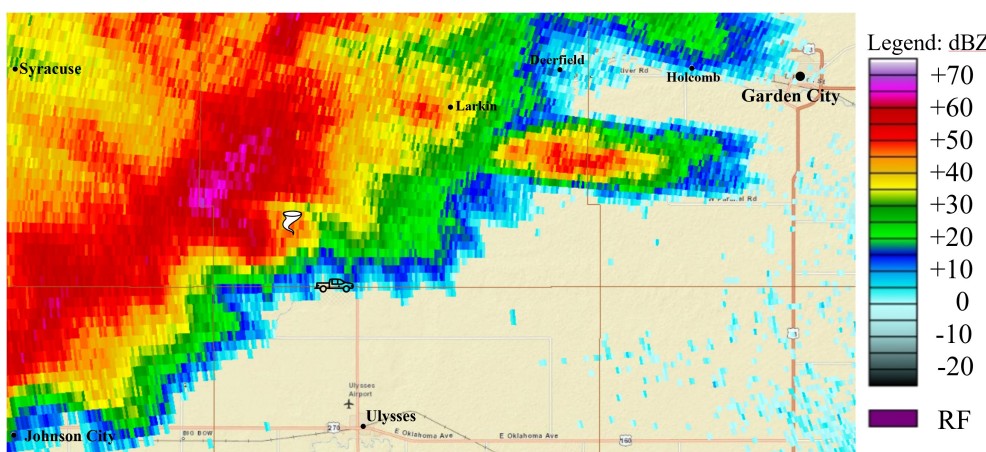

**Figure B1.** KDDC (Dodge City, KS) reflectivity measurement at 22 May 2020, 0017 UTC with elevation of the scan = 2590 ft (lowest possible base scan). The tornado location is indicated via the tornado icon and the stormchaser region of intercept via the truck icon.

**B2 Radial Velocity Measurements**

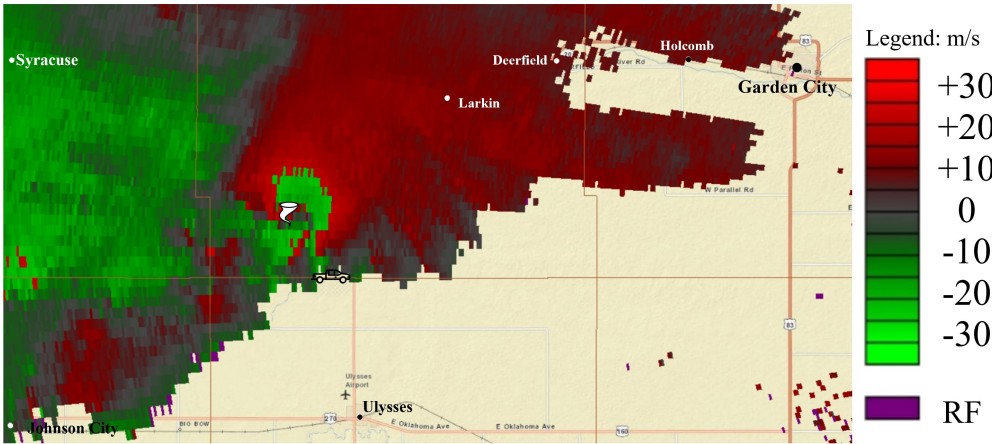

**Figure B2.** KDDC (Dodge City, KS) radial velocity measurement at 22 May 2020, 0017 UTC with elevation of the scan = 2590 ft (lowest possible base scan). The tornado location is indicated via the tornado icon and the stormchaser region of intercept via the truck icon.





*Author contributions.* B. White designed and fabricated the equipment and implemented the processing and analysis, I. Faruque provided the analysis approach, and B. Elbing provided the experimental context. All authors contributed to interpreting the results and writing.

*Competing interests.* None.

*Acknowledgements.* The authors are grateful to Val and Amy Castor for hosting the GLINDA unit on their vehicle during the 2020 and 2021 storm seasons, and to Prof. Jamey Jacob for providing the calibration noise source. This work was funded in part by the National Oceanic and Atmospheric Administration (NOAA) under grants NA18OAR4590307 and NA19OAR4590340. This paper includes experimental data. These data and related items of information have not been formally disseminated by NOAA, and do not represent any agency determination,
view, or policy.





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
