# Peer review of "INFRASOUND MEASUREMENT SYSTEM FOR REAL-TIME IN-SITU TORNADO MEASUREMENTS"

_Atmospheric Measurement Techniques, 2021_

## Author Comment (AC1)

Authors' Response to Report #1

The authors would like to thank the reviewer for their time, dedicated attention to detail, and thorough review of the manuscript. Below are the reviewer's specific comments retyped in *black and the italicized* style with corresponding response indicated in **blue and bold**. Once again, the authors want to offer their sincere gratitude to the reviewer for their contribution to ensuring this paper's quality and relevancy.

- Authors Brandon C. White, Brian R. Elbing, and Imraan A. Faruque

*Abstract* = "*Hasproviding*" – *one word*?

This line has been revised as "GLINDA has been deployed with storm chasers beginning in May 2020 and has provided real-time automated monitoring of spectrum and peak detection."

L48: Why is there a tilde over 1 Hz?

The line has been updated from "Bedard (2005) showed that ~1 Hz infrasound emissions followed the available radar observations associated with a tornado." to "Bedard (2005) showed that infrasound emissions of  $\approx$  1 Hz followed the available radar observations associated with a tornado."

*P6L110: What are the deployment conditions during the measurements? Are there any potential acoustic interference from other equipment or environment noise (e.g. weather radar, truck engine, traffic or nearby industrial)?*

The external deployment conditions vary spatially as the chasing vehicle moves during intercept, therefore noise from elements such as traffic, industrial, and/or natural sources may be represented differently across the time series data while intercepting. In the measurements presented for the Lakin tornado event, these external sources are expected to be of mitigated concern due to the geographic setting being a largely open field/farm area (reducing industrial noise and traffic).

As opposed to the spatially distributed sources, truck and other sensor noise is in constant relative position to the GLINDA system during all time of operation. While many sources of measurement noise (inverter 60 Hz signal, human speech, etc) are above the frequency band of interest, any signals which may be regularly seen in operation over 1-100 Hz can be considered in noise floor measurements. This is a driving motivator to the contents of Fig. 12 where measurements during the life of the tornado are compared to 1 hr before and after tornadogenesis. The apparent rise over tornado-infrasound frequencies is above the noise floor of all internal noise (represented by the blue, Pre-Event curve).

*P7L137: What are the test conditions of the measurement shown in the spectrogram of Fig.5? The Figure should be introduced in more detail, such as meaning of Fres and Tres in the title.*

The paragraph has been revised to improve clarity. It now reads, "The primary visualizations of the data are a spectrogram displaying near real time frequency decompositions and a maps API displaying the location of the storm chasing unit via GPS. Figure 5 shows a demonstration of these visualization capabilities with simulated inputs. In the spectrogram,  $F_{res}$  and  $T_{res}$  represent the size of the frequency and time bins respectively.

*L164: "fixed frequency domain resolution over the 0 to Nyquist frequency range." I'm not sure I understand this sentence.*

This statement has been revised as follows: "Traditionally the frequency decomposition of a timedomain signal is performed using a Fast Fourier transform (FFT) which returns a frequency domain representation of the data with linearly-spaced frequency points over the frequency band  $f_{FFT} \in [0, f_s/2]$ , where  $f_s/2$  is commonly known as the Nyquist frequency." *P9L183: Fig. 6 is missing(?). Only a caption appears.*

Figure 6 has been appropriately compiled into the document and now appears above its appropriate caption. It is additionally included here for reference.

P10L196: The storm chaser "intercepted" the tornado before its generation. Is this sentence means the GLINDA system captured the characteristic infrasound of the tornado before its touchdown? If so, could the author provide the accurate time difference (instead of "shortly") between acoustic interception and physical tornadic touchdown? This is very valuable data.

During the lead-up to tornadogenesis, the storm chasing vehicle was en route to the site of rotation. In general, the "intercept" or time window with chasers at the identified measurement proximity to the site of the tornado occurred approximately 2-5 minutes before the tornado touched down. The impacts of spatial rate of change on propagation of infrasonic signals and its measurement are not well known at this time and the subject of ongoing study which includes notable uncertainty. Once intercept of the tornado was established by the chasers, the vehicle remained largely stationary with panned video coverage resulting in expected reduced process noise.

This line has been revised as "The storm chasers, equipped with GLINDA, arrived to the intercepting location for the tornadic storm system approximately 2-5 minutes prior to tornadogenesis. The intercepting storm chasers were located approximately 4 km SSE of the tornado during tornadogenesis."

P18L293: In conclusion, the term GLINDA has been already defined in previous P3L58, which is a different definition.

The conclusion has been updated to reference the full name of "Ground-based Local INfrasound Data Acquisition (GLINDA) system" as in earlier text.

P18L307: The "past observation" may need reference.

A reference to the earlier discussion of section 2 is included as "The spectral content shows an elevated signal during the tornado spanning 10 to 15 Hz, consistent with past observations of small tornadoes as described in section 2."

*Figure captions, figure caption, figure property, math equation, and abbreviation. Examples are listed below.*

**P3L72: Usually, there is no one-sentence paragraph for academic writing.**

This introductory sentence has been expanded upon to further clarify the content of the following paragraphs in the section. The introduction now reads "In this section, system design goals are identified, and hardware components, computational platforms, and data handling for collection and retention prior to analysis are discussed. Additionally, calibration procedures over the specific range of frequencies of interests are presented for the unit."

P5L94: Figure caption usually ends with a period.

A period has been added to end the caption.

P9L172: Reference of equation should be Eq. 2 or Eqn. 2.

This line now reads, "The CZT, defined in Eq. (2), ..."

P9L179: Equation 2 does not have a punctuation.

Equation 2 has been reviewed

6.P12L225: Space is needed between number and unit.

Corrected from "500Hz" to "500 Hz"

P15: Fig. 14 axis's labels need revision.

Font size has been reduced to be more in line with manuscript font size as seen below

---

## Author Comment (AC2)

The authors of the manuscript appreciate the reviewer taking time to read and provide commentary on the paper to assist in its improved quality. The inclusion of meteorological perspective from the reviewer are invaluable to fully shaping the content of the paper. Below, the reviewer's comments are reproduced in the *black, italics style* with author response provided below in the **bold, blue style.** Once again, the authors would like to offer their appreciation for the review's time and assistance.

- Authors Brandon C. White, Brian R. Elbing, and Imraan A. Faruque

|
 |
 |  |
|------|------|--|
|
 |
 |  |
|      |      |  |
|      |      |  |
|      |      |  |
|      |      |  |
|      |      |  |
|
 |
 |  |
|
 |
 |  |
|      |      |  |

Near line 20. I think the bigger issue in the Southeast and elsewhere is the radar horizon. There aren't too many places where the base elevations are blocked.

This line has been amended to, "Many fatalities occur in the southeast United States due, in part, to hilly terrain limiting line of-sight measurements (such as radar) and radar horizon interactions".

Near line 20. I suggest breaking the multiple citations into those that pertain generally to infrasound studies, and those that suggest there is specific tornado structure/dynamics information.

All of these references have tied infrasound to tornadoes, ie, from the literature: Bedard 2004b: "might have potential for tornado detection and warning." Frazier 2014: Frazier et al. (2014) examined high-fidelity acoustic recordings covering the frequency range from 0.2 to 500Hz from three tornadoes in Oklahoma. (from Elbing 2019) Goudea 2018: "Analysis of these bearing estimates revealed a trend in which storms often depicted infrasonic emissions during the time preceding tornadogenesis, followed by a rapid increase in emission intensity as tornadic activity commenced, and a subsequent weakening of the acoustic signal after tornado dissipation."

Elbing 2019: "in the 5–50Hz band the infrasound was independent of wind speed with a bearing angle that was consistent with the movement of the storm core that produced the tornado. During the tornado, a 75dB peak formed at 8.3Hz, which was 18dB above."

Near line 21. Table 1 does not provide information on how tornadoes can be predicted or understood.

The line in the text has been modified to delineate our suggested potential future state from the table which shows comparison of properties.

The line has been modified from "The long propagation range of infrasound, coupled with the omnidirectional, continuous coverage provided by relatively inexpensive infrasound microphones could provide a significant improvement in our ability to detect, track, and ultimately predict and understand tornadic phenomena, as summarized in Table 1." to "Table 1 provides a comparison between aspects of infrasound measurements and the respective properties as observed in the tornado dynamics and radar measurement. The long propagation range of infrasound, coupled with the omnidirectional, continuous coverage provided by relatively inexpensive infrasound microphones could provide a significant improvement in our ability to detect, track, and ultimately predict and understand tornadic phenomena."

Near line 21. Instead of "decentralized" would "mobile" be more appropriate? What does "provide widespread realtime infrasound coverage near tornado-bases without additional cost to the end user" mean? I'm puzzled.

This line has been amended to, "These are the first tools available to the public that are applicable to mobile deployment and would provide widespread real-time infrasound coverage near the bases of tornadoes without additional cost to the end user."

Near line 40 "the details of the association technique are not included...". I'm glad you made this disclaimer; it makes this study of little value, at least as reported in the literature.

**Response N/A**

Near line 40. Did Dunn track tornadoes, or tornadic storms, or mesocyclones? Specifically, did the signal begin and end at the start/stop times of the tornadoes?

Dunn reports on four tornado-producing storms and the signal "was initially observed 30 min before the funnel was reported on the ground." More detail has been included to this line.

Love the acronym GLINDA :-)

**Response N/A**

Near line 76... check the "accuracy" of NWS tornado reports. They may report to 10 m significant digits in lat/lon, but I'm guessing the start stop points are rarely known to better than 100 m or worse.

The line has been updated to reflect that NWS "does not guarantee the accuracy or validity of the information", and that comparison is intended to represent the accuracy of the system's measurement.

The line now reads, "... (b) positioning resolution under 10 m to provide comparable or better precision to current NOAA-reported tornado coordinates (the NWS does not provide uncertainty or accuracy estimates for this number that would quantify accuracy), ..."

"Accordingly, the NWS does not guarantee the accuracy or validity of the information," from https://www.nws.noaa.gov/directives/sym/pd01016005curr.pdf#:~:text=Accordingly%2C%20the%20 NWS%20does%20not%20guarantee%20the%20accuracy,requiring%20additional%20information%20s hould%20contact%20that%20source%20directly.

Near line 194... personal pet peeve... I really don't like the term "touch down" for tornadoes. Tornadoes just don't do this. Prior to formation of a tornado, there is a vortex extending to within meters of the ground, with the vortex lines extending horizontally outward from there. Depending on where the stretching is most intense, the vortex first becomes "tornadic" near the ground, or a variety of other heights. A "tornado formed" is much to be preferred. This might have implications for your work... a vortex can be quite strong at various heights prior to the condensation/debris that characterizes a tornado, so the begin/end of the visible manifestation, or even the damage, should not be expected to correspond exactly to your infrasound signals (unless, of course, those only occur when damage is being done or the vorticity reaches a threshold magnitude).

The line has been amended to "The tornado formed at 0011 UTC at coordinates..."

Near 195-196 the precision of lon, lat, length, and width are much beyond what the NWS accomplishes in reality. I fault the NWS for using this precision in their reports.

The line has been updated to reflect "NWS-reported" coordinates which calls back to the previous commentary on their lack of accuracy/uncertainty values provided.

195. Tornadogenesis is a process, not an event.

This line has been amended to refer to tornado formation rather than tornadogenesis, "... arrived to the intercepting location for the tornadic storm system approximately 2-5 minutes prior to tornado formation."

205. "weak rotation" is problematic for your study, is it not? I.e., it makes refuting the hypothesis of infrasound from the mesocyclone more problematic. Figure 11 and Figure 12... this is about as deeply as I can delve into the signal analysis work. There are differences, clearly. But there simply cannot be conclusions drawn from two events that are "different", except that "differences can happen".

The authors do not view the weak rotation exhibited by the hail event to be problematic for the study. Rather, the authors offer the comparison of the two events simply to note that the presence of rotation observed in the hail event did not produce a similar spike in infrasonic signal content (Figure 13) relative to pre and post event signals like is observed following tornado formation in Figure 12. This is not intended to establish firm conclusion, however, to note that the observation is in line with previous literature associating the infrasound to tornado formation specifically.

And with that latter conclusion, of course, it is setting the stage for much more needed work across a statistically useful sample size. I'm not sure what this will be... additional cases will shed light. I have to believe that perhaps dozens of cases will be needed to see if the signal is really unique in tornadoes.

The authors agree with the assessment that further investigation to confirm the association of infrasonic signal and tornado formation is needed – including multiple measurements of tornadic events. The system we present in this paper is suggested for making such observations as it has been demonstrated in a relevant, real-world context.

The amount of processing needed to extract these fairly small signals (is that a fair characterization?) makes me really worry that the analysis was tuned to extract the strongest possible result.

The methodology presented requires very little tuning or specialized processing to return the results in this paper. The selection of windowing parameters and implementation of the frequency decomposition are largely dictated by the frequency band of interest which can be considered static from case to case (as we maintain focus on the frequency band [0.1, 250] Hz in this paper). The sensitivity of the analysis for the transfer function approximations is presented in Table 5 with note that the values returned are highly consistent across the Monte Carlo runs.

This, of course, is pretty much the standard approach with rare phenomena... you want to know if there is any potential for discrimination. Again, just be clear that these uncertainties are part and parcel of limited data sets of rare phenomena. There seems to be a general perception in the weather community that infrasound is going to provide information that is valuable for tornado anticipation. This may be the goal, and it may be what is communicated via the popular media, but this paper does not bolster that case. I'm not aware of formal literature that clearly does make that case.

Section 5's introduction has been revised to clarify that the observations presented are with intent to demonstrate the processing methodology available through the measurement system rather than to conclude infrasound-association to tornado formation.

Please be clear about what is the eventual goal/hope of this research, and what is actually known. Two cases that exhibit somewhat different signals, obtained at close range, point the way to additional useful observations. They are simply not evidence of the utility of infrasound operationally. It must be shown that the signal is consistent, and that it occurs even when humans are not able to see the tornado visually or via radar signatures. And perhaps most important, I hope you will find a signal of the processes occurring in the 20 min prior to tornado formation. This is the time period when warnings are pretty bad. After tornado formation, it's pretty rare that the NWS is completely unaware of the presence of a tornado.

This paper presents and demonstrates the capabilities of the GLINDA system to make infrasound observations at close range to tornadic events which can inform continuing work on the potential relevance of infrasound to tornado detection and tracking. Although the paper presents example observations indicating an SPL rise for the tornadic case, these observations are intended for demonstrating the capabilities of the system and processing techniques rather than to be interpreted as a firm conclusion of the efficacy of infrasound-based tornado detection. The further collection of data with the GLINDA system would allow for interrogation of signal onset timing and association to source.